# Real-to-Sim Grasp: Rethinking the Gap between Simulation and Real World in Grasp Detection

Jia-Feng Cai [1], Zibo Chen [1], Xiao-Ming Wu [1], Jian-Jian Jiang [1], Yi-Lin Wei [1], Wei-Shi Zheng [1,2†]

[1] School of Computer Science and Engineering, Sun Yat-sen University, China

[2] Key Laboratory of Machine Intelligence and Advanced Computing, Ministry of Education, China

{caijf23, chenzb8, wuxm65, jiangjj35, weiylin5}@mail2.sysu.edu.cn    wszheng@ieee.org

**Abstract:** For 6-DoF grasp detection, simulated data is expandable to train more powerful model, but it faces the challenge of the large gap between simulation and real world. Previous works bridge this gap with a sim-to-real way[1]. However, this way explicitly or implicitly forces the simulated data to adapt to the noisy real data when training grasp detectors, where the positional drift and structural distortion within the camera noise will harm the grasp learning. In this work, we propose a **R**eal-to-**S**im framework for 6-DoF **Grasp** detection, named R2SGrasp, with the key insight of bridging this gap in a real-to-sim way[2], which directly bypasses the camera noise in grasp detector training through an inference-time real-to-sim adaption. To achieve this real-to-sim adaptation, our R2SGrasp designs the Real-to-Sim Data Repairer (R2SRepairer) to mitigate the camera noise of real depth maps in data-level, and the Real-to-Sim Feature Enhancer (R2SEnhancer) to enhance real features with precise simulated geometric primitives in feature-level. To endow our framework with the generalization ability, we construct a large-scale simulated dataset cost-efficiently to train our grasp detector, which includes 64,000 RGB-D images with 14.4 million grasp annotations. Sufficient experiments show that R2SGrasp is powerful and our real-to-sim perspective is effective. The real-world experiments further show great generalization ability of R2SGrasp. Project page is available on https://isee-laboratory.github.io/R2SGrasp.

**Keywords:** Grasp pose detection, simulated datasets, sim-to-real

## 1 Introduction

Grasping objects in unstructured environment is fundamental for robots designed to accomplish various complex tasks[1, 2, 3]. Previous works [4, 5, 6, 7, 8, 9, 10] utilize real-world data to achieve impressive performance. However, collecting large-scale datasets in real world is still challenging, which limits the improvement of the grasp and generalization abilities. To address this problem, simulated data provides a feasible alternative, but the grasp performance trained in simulation is hindered by the large gap between simulation and real world when applying to real world scenarios.

There exist a few methods attempting to bridge this gap, which use domain randomization [6, 11, 12] or domain adaptation [13, 14, 15, 16, 17] to achieve it. The former adopts various randomized conditions to make the simulated data closer to reality, while the latter narrows the gap by aligning feature distribution between two domains. However, they both include a sim-to-real way that adapts the simulated distribution to the real-world one, which explicitly or implicitly introduces the camera noise in real data when training the grasp detector. This noise appears as positional drift and structural distortion as shown in Figure 1 (a), disrupting the training process of the grasping detector that

---

[†]Corresponding author.

[1]"sim-to-real way" refers to making simulated data or features more similar to real-world data or features.

[2]"real-to-sim way" involves making real-world data or features more similar to simulated data or features. Our real-to-sim process bypasses the camera noise in grasp detector training and obtains great performance.

8th Conference on Robot Learning (CoRL 2024), Munich, Germany.

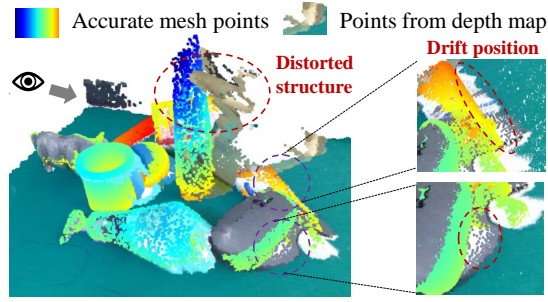
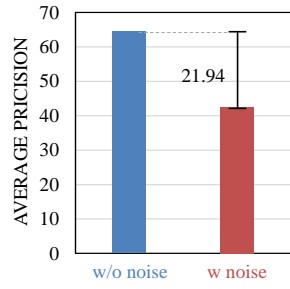

(a) Real-world point clouds with mesh (b) Upper bound of grasp performance

Figure 1: Illustrations of problems related to the gap between simulation and real world. Figure (a) shows mixed point clouds, including single-view point clouds of the scene and point clouds sampled from accurate object meshes. There are positional drift and structural distortion in real-world single-view point clouds which are caused by camera noise in real data. Figure (b) depicts that the camera noise disrupts the training of the grasp detector, as the average precision of the grasp detector trained in point clouds with real-world noise is lower than that trained in noiseless point clouds.

is highly sensitive to variance in 3D space. To demonstrate the negative effect of camera noise, we compare the upper bound of grasp performance of the grasp detector trained on noiseless data and on noisy data as depicted in Figure 1 (b). The performance of grasp detector trained in data with real-world noise is lower than that of trained in noiseless data, which suggests that the camera noise introduced by the sim-to-real way will disturb the learning of grasping skills.

To this end, we propose a novel **R**eal-to-**S**im framework for 6-DoF **Grasp** detection, named R2SGrasp, to bridge the gap between simulation and real world in a real-to-sim way. **This way adapts real-world distribution to the simulated one in inference phase to leverage the precise grasping skills learned from the simulation, which bypasses the camera noise in detector training and ensures robust performance.** Specifically, a grasp detector is trained with noiseless simulation data to attain accurate grasping capability. During inference, we achieve real-to-sim adaptation by using a Real-to-Sim Repairer (R2SReparier) to preprocess the depth map so as to mitigate positional drift and structural deformation, along with a Real-to-Sim Feature Enhancer (R2SEnhancer) to enhance real-world local features with simulated structural features. Benefiting from our two novel modules, R2SGrasp implements the real-to-sim way in data and feature levels to achieve higher performance.

Moreover, benefiting from our real-to-sim perspective, our grasp detector only requires training in simulated data, which endows our framework with the expandable generalization ability by scaling up the simulated data cost-effectively. Therefore, we build a large-scale simulated dataset, named R2Sim, including 256 daily objects, 500 cluttered scenes and 64,000 RGB-D images with total 14.4 million grasp annotations. To improve the efficiency of training on such a large-scale simulated data, we propose a simplified annotation strategy, resulting in a reduction on training time by 73%.

We conduct extensive experiments on the GraspNet-1Billion dataset [5] to verify the effectiveness of our R2SGrasp. R2SGrasp achieves superior performance in real world only with simulated data, surpassing methods of using annotated real-world data. Moreover, we further evaluate R2SGrasp through many real-world grasping experiments, verifying the great generalization of our framework.

## 2 Related Work

**Sim-to-real transfer in grasp detection**. The gap between simulation and reality is a significant problem in robotics grasping. For those RGB-based grasping methods, most works utilize advanced image processing techniques, such as adversarial training [13, 14], GAN-based methods [17, 18] or teacher-stundent models [15] to bridge the gap between simulated and real RGB images or features. We focus on the transfer of point cloud-based grasping methods from simulation to reality. Fang et al. [6] adds Gaussian noise to the point cloud for training. Zheng et al. [16] aligns the features

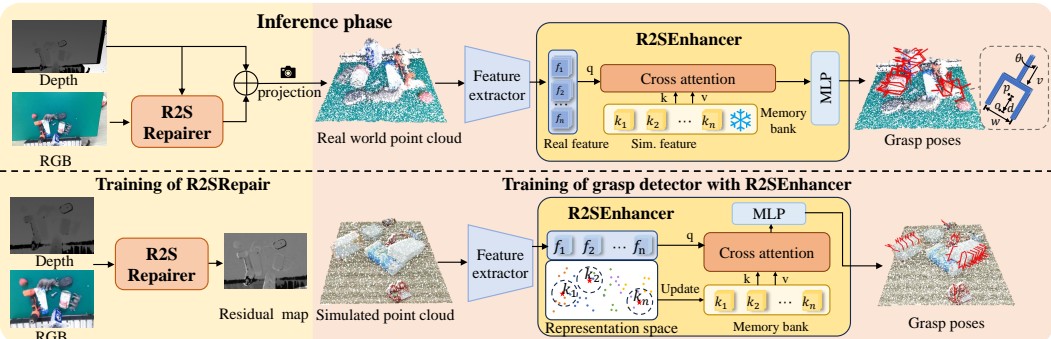

Figure 2: Overview of R2SGrasp framework. In inference phase, the Real-to-Sim Data Repairer (R2SRepairer) repairs depth map from RGB-D input, then a feature extractor extracts local features from the single-view point cloud which is transformed from the repaired depth map. Then Real-to-Sim Feature Enhancer (R2SEnhancer) enhances the real features using the stored simulated structural features and finally predicts the grasp poses. In training phase, we train the R2SRepairer on twin datasets and train the grasp detector with R2SEnhancer on our R2Sim dataset.

between the real and simulated domain through adversarial training. The above sim-to-real methods in point clouds enable the grasping model to handle the noise at the data and feature levels, but the introduction of noise during training will reduce the upper bound of the performance of the grasp detector. In contrast, our method based on point clouds makes the real data and features closer to simulation ones.

**Grasp datasets in simulation**. Simulation datasets [18, 19, 20] are the most commonly used data for robot learning. Previous works [21, 22, 23, 24] propose the datasets that define grasp poses within the continuous grasp sample space, which have insufficient annotations, challenging to cover the entire sample space. Recently, [5] uses discrete annotations to fully cover the entire sample sample so as to learn the robustness of grasp abilities. Following that, we decouple the grasp pose parameters and discretize them at fixed intervals within the parameter space. Our annotations are rich enough to cover the entire discrete parameter space. Due to the demand for large-scale simulation training, we also propose a simplified annotation method to enhance training efficiency. Compared to previous simulated datasets, our dataset employs a discrete representation of grasp poses, and provide more dense grasp annotations that cover the entire sample space, which enables our model to achieve great generalization ability.

## 3 R2SGrasp: A Real-to-Sim Framework for 6-DoF Grasp detection

### 3.1 Task Definition

We first describe the task of 6-DoF grasp detection. Given the single-view RGB-D image $I_{rgbd} \in \mathbb{R}^{H \times W \times 4}$, the goal of grasp detector is to predict a set of accurate grasp poses in cluttered scenes. The 6-DoF grasp pose can be denoted as $G = [R, T, w]$, where $R \in SO(3)$ is the rotation matrix, $T \in \mathbb{R}^3$ is the translation and $w$ is the gripper width. We follow [5] to decouple $R$ into approaching vector $v$ and in-plane rotation $\theta$, $T$ into grasp point $p$ and approaching distance d from grasp points to grasp origin $o$, as shown in Figure 2. In this work, we aims to bridge the gap between simulation and real world in 6-DoF grasp detection, where we will train our model in simulation and test it in real world data for employment.

### 3.2 Why should we use real-to-sim adaptation?

Previous works employ the sim-to-real way to bridge the gap between simulation and real world. However, they explicitly or implicitly introduce the camera noise in real data when training the grasp detector. The positional drift and structural distortion caused by camera noise disrupt the training

process of the grasp detector, limiting the grasp performance, as depicted in the right hand side of inference phase in Figure 1 (c).

To solve the above problem, we propose a novel real-to-sim perspective that bridges the gap between simulation and real world in a real-to-sim way. This way is able to train a robust grasp detector on noiseless simulated data to avoid the interference of noise in grasping skills learning, thereby achieving stronger grasping capabilities. Although this grasp detector performs exceptionally well on noiseless simulated data, its performance degrades on real data due to the presence of noise. To apply the precise grasping ability learned from simulation to the real world, we consider adapting the real world data and feature to the simulated ones. Specifically, we introduce the R2SGrasp framework, including a Real-to-Sim Data Repairer (R2SRepairer) and a Real-to-Sim Feature Enhancer (R2SEnhancer) to achieve real-to-sim adaptation at data and feature level respectively. The overview of R2SGrasp is shown in Figure 2. Except the two modules, our framework also consist of a grasp detector to generate the aggregated features of the grasp points and a MLP to output the grasp widths and grasp scores for each grasp candidate along the approach direction, following [8]. We adopt the same loss function as [8] to train our grasp detector. More details about our grasp detector and loss function can be seen in Appendix A.2. How the two modules adapt the real world data and feature to the simulated ones will be detailed below.

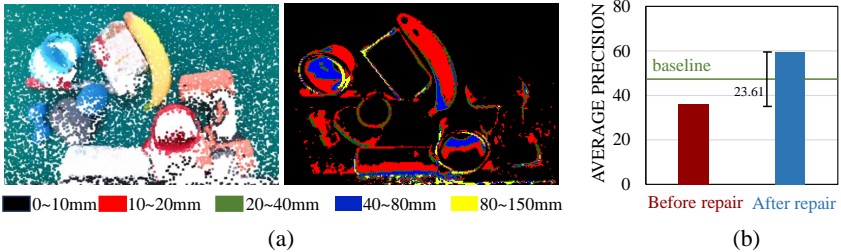

Figure 3: Illustrations on the impact of camera noise. (a) shows the point cloud and noise map, where different colors in the noise map represent different noise amplitude ranges, with amplitude measured in millimeters. (b) shows the performance difference before and after depth map repair using ground truth, and the green line presents the real-world training performance.

### 3.3 Real-to-Sim Data Repairer

**Impact of camera noise.** We find that the gap between simulated and real domain primarily stems from camera noise in real data which appears as the positional drift and structural deformation in point clouds. As shown in Figure 3 (a), there is severe noise in depth map captured from the real world. To explore the impact of this noise, we conduct a verified experiment that simply replacing the severe noise regions in the real depth map with accurate depth values to repair the structure. As shown in Figure 3 (b), the average precision increases by 23.61%, even surpassing the baseline [8] whose model trained on real dataset. The results suggest that if we can mitigate this noise and repair the structure, the grasp performance can be improved significantly.

**Mitigating positional drift and structural deformation.** We propose the real-to-sim data repairer (R2SRepairer) to mitigate positional drift and structural deformation by reducing the noise amplitude using an encoder-decoder architecture. It takes RGB-D images $I_{rgbd} \in \mathbb{R}^{H \times W \times 4}$ as input and outputs residual map $I_r \in \mathbb{R}^{H \times W}$ . The repaired depth map $\hat{I}_d$ can be obtained as Eq. (1):

$$\hat{I}_d(i,j) = I_d(i,j) + I_r(i,j), \tag{1}$$

$$I_n^*(i,j) = I_d^s(i,j) - I_n^r(i,j). \tag{2}$$

To obtain supervision data $I_n^*$, we use the object models and object poses from GraspNet-1Billion [5] to efficiently generate the simulated data paired with the real ones in Blenderproc [25]. By setting the camera position in the simulation environment to match the real camera pose, we can render simulated depth maps that are identical to the real depth maps from GraspNet-1Billion. $I_n^*$

is calculated as Eq.(2), where $I_d^s$ and $I_n^r$ are paired simulated and real-world depth maps. The R2SRepairer is trained with Smooth L1 loss.

### 3.4 Real-to-Sim Feature Enhancer

To further bridge the gap between the simulation and real world, R2SEnhancer conducts the real-to-sim adaptation at feature level by using the precise structural features to enhance the real-world local features based on feature similarity. As shown in Figure 2, memory bank in R2SEnhancer accumulates diverse geometric structure information through clustering during training with simulated data. During inference, this simulated structure information is used to enhance the features of real structures, thereby improving the perception of our grasp detector in real geometric structures.

**Building the memory bank.** Memory bank $M = \{k_j\}_{j=1}^K$ stores $K$ simulated structure features $k_j$ during training. At first, we initialize $M$ by sampling from the normal distribution. Given the local features $F_l^s = \{f_i\}_{i=1}^{N_s}$ of each batch extracted in simulated data when training, we calculate the distance between local features and stored features based on cosine similarity, and assign each local feature to the closest stored feature. We then update the stored features with the average of all assigned feature clusters using momentum update as Eq.(3):

$$k_j = \alpha k_j + (1 - \alpha)\bar{f}_j, \tag{3}$$

where $\bar{f}_j$ is the average feature of the cluster corresponding to $k_j$, and $\alpha \in [0, 1]$ is the momentum update parameter. After that, we only keep the centers of clutters for the next-batch training.

**Real-world feature enhancement.** After training with extensive simulated data, the memory bank stores a diverse set of structural features that represent common geometric structures. We retrieve similar simulated structural features to enhance the real features through cross-attention mechanism, where the local features $F_l^r = \{f_i\}_{i=1}^{N_s}$ extracted from real-world point clouds serve as the queries, and the simulated structural features stored in the memory bank serve as the keys and values. The enhanced real features are obtained using Eq.(4).

$$\hat{F}_l^r = \rho((F_l^r W^q)(MW^k)^T / \sqrt{D_m})(MW^v) + F_l^r, \tag{4}$$

where $\rho$ denotes the Softmax function, $W^q, W^k, W^v \in \mathbf{R}^{C \times D_m}$ are learnable encoding matrices for query, key and value in cross-attention mechanism.

## 4 R2Sim Dataset

**Motivation of the R2Sim Dataset.** Thanks to our real-to-sim perspective, our grasp detector only needs simulated data for training. This endows our framework the expandable generalization ability by scaling up the simulated data cost-effectively. Towards this end, we construct a large-scale dataset named R2Sim in Blenderproc [25], including 256 daily household objects selected from GSO [26] and GraspNet-1Billion [5], 500 cluttered desktop scenes, 64,000 RGB-D images taken from different views and approximately 14.4 million grasp annotations. Each frame is also annotated with object segmentation map, object poses, camera pose, graspness heatmap [8] and so on. More information about our R2Sim is available in Appendix A.3.

**Scene generation.** We provide an automated process for generating scene data with various randomization techniques to enhance the diversity, such as rich background textures, diverse lighting conditions, cluttered table scenes, and various random camera views. This process automatically generates RGB-D images, segmentation masks, 6D object poses, and camera poses, providing rich training data for the grasp detector. More details are available in the Appendix A.3.3.

**Grasp pose annotation for efficient training.** Given the object meshes, we use a discrete dense annotation method [5] for object grasp pose labeling, where the grasp points are densely and uniformly distributed on the object surface, with each grasp point having 300 approach directions and 48 grasp candidates per direction. Scene grasp pose annotations can be obtained by projecting the grasp annotations of objects according to their 6-DoF poses. As the data scale expands, the dense

annotations significantly increase training burden, including prolonged data loading time and excessive memory consumption. To improve training efficiency, we simplify the grasp pose annotations by reducing the number of negative samples. Specifically, we first filter out grasp points that do not have successful grasp candidates. Then, for the retained grasp points, we select the top 60 approaching vectors from the original predefined 300 vectors based on the proportion of successful grasp candidates. Benefiting from our simplification, we filter out many invalid grasp annotations, which improves the training efficiency and is more suitable for large-scale simulation data training. More details and further experimental validation are illustrated in Appendix A.3.4.

## 5  Experimental Results

### 5.1  Experimental setup and details

**Datasets.** We use R2Sim dataset to train the grasp detector and use twin datasets based on GraspNet-1Billion [5] to train our R2SRepairer. The grasping performance is validated using the test set from GraspNet-1Billion. GraspNet-1Billion is a real-world dataset comprising 190 scenes, with 100 scenes allocated for training and 90 for testing. The test set is further divided into three subsets: seen, similar, and novel, where each category contains 30 scenes.

**Metrics.** We employ $AP_\mu$ and AP as the evaluation metric for grasp performance, the metrics introduced by [5]. $AP_\mu$ represents the average precision of the top 50 grasp poses in terms of grasp scores under different friction coefficient $\mu$. AP denotes the mean of $AP_\mu$.

**Implementation Details.** The backbone of R2SRepairer is implemented using a UNet [27] architecture with a ResNet34 [28] encoder, outputting feature vectors with 32 channels. The feature extractor in our grasp detector adopts the cascaded graspness model in GSNet [8] with cylinder grouping operation. The cross-attention mechanism consists of four heads, with a model dimension of 256. The storage capacity of the memory bank $K = 120$, with the momentum update parameter $\alpha = 0.999$. Additionally, R2SRepairer is trained with a learning rate of 0.001, using the AdamW optimizer and a batch size of 3 for 30 epochs. And the grasp detector is trained with a learning rate of 0.001, using the Adam optimizer and a batch size of 4 for 30 epochs.

### 5.2  Experiments on GraspNet-1Biilion

**Comparison with sim-to-real methods.** We first compare our method with current sim-to-real approaches as shown in Table 1 (the methods without using labeled real data). Source-only trains the grasp detector in simulted data without any sim-to-real techniques. ADD-Noise [6] adds Gaussian noise to the simulated point cloud. Global-DA [29] and Local-DA [29] utilize adversarial domain adaptation methods to align the distributions of the simulated and real domains at global and local feature levels respectively. Note that the above methods utilize the same grasp detector [8] as ours for fair comparison. It is evident that our method significantly surpasses others by 9.19/18.64, 3.65/7.62, 5.07/10.14 AP at least in the seen, similar, and novel categories respectively. Additionally, the results of previous methods have little improvement, even decline on grasp performance. We think the reason is that these works explicitly or implicitly introduce the camera noise in real data when training the grasp detector, which disturbs the learning of grasping skills.

**Comparison with methods trained in real world.** To further illustrate the advantages of R2SGrasp, we also compare our method with others [5, 9, 8, 7, 4] trained on real-world datasets, as shown in Table 1 (the methods using labeled real data). The performance of our method is close to that of models trained on real data, and it even surpasses current SOTA methods in most metrics. Specifically, compared to GSNet [8], our method outperforms it by 4.93, 7.08, and 5.73 AP in the seen, similar, and novel categories on Kinect data. The results demonstrate that our R2SGrasp can leverage the precise grasping skills learned from the simulation in the real world.

**Ablation Study.** We conduct ablation study to validate the effectiveness of each component in our method as shown in Table 2. Line 1 is the control group, which is trained on real-world data. When

| Using labeled real data | Methods | Seen | | | Similar | | | Novel | | |
|---|---|---|---|---|---|---|---|---|---|---|
| | | AP | $AP_{0.8}$ | $AP_{0.4}$ | AP | $AP_{0.8}$ | $AP_{0.4}$ | AP | $AP_{0.8}$ | $AP_{0.4}$ |
| ✓ | Graspnet-baseline[5] | 27.56/29.88 | 33.43/36.19 | 16.95/19.31 | 26.11/27.84 | 34.18/33.19 | 14.23/16.62 | 10.55/11.51 | 11.25/12.92 | 3.98/3.56 |
| | TransGrasp[9] | 39.81/35.97 | 47.54/41.69 | 36.42/31.86 | 29.32/29.71 | 34.80/35.67 | 25.19/24.19 | 13.83/11.41 | 17.11/14.42 | 7.67/5.84 |
| | GSNet[8] | **65.70**/61.19 | **76.25**/71.46 | **61.08**/56.04 | 53.75/47.39 | 65.04/56.78 | 45.97/40.43 | 23.98/19.01 | 29.93/23.73 | 14.05/10.60 |
| | HGGD[7] | 59.36/60.26 | - | - | 51.20/48.59 | - | - | 22.17/18.43 | - | - |
| | GRE-Grasp[4] | -/65.19 | -/75.37 | -/**59.22** | -/54.09 | -/**64.25** | -/47.14 | -/22.29 | -/27.69 | -/13.53 |
| ✗ | Source-only | 54.85/46.72 | 66.65/55.67 | 44.26/38.92 | 53.28/46.85 | 64.81/56.41 | 44.63/39.22 | 21.23/14.60 | 26.84/18.58 | 10.68/7.39 |
| | ADD-Noise[6] | 55.52/47.48 | 67.24/56.19 | 45.57/40.11 | 53.11/45.05 | 64.32/54.16 | 45.4038.07 | 21.24/14.49 | 26.98/18.47 | 10.51/7.35 |
| | Global-DA[29] | 47.16/42.28 | 57.21/50.89 | 38.22/34.22 | 44.15/40.66 | 54.44/49.78 | 35.47/32.52 | 17.66/12.27 | 22.39/15.57 | 8.62/6.18 |
| | Local-DA[29] | 53.21/45.99 | 63.98/54.58 | 44.07/38.56 | 50.88/43.77 | 61.07/52.51 | 44.43/36.93 | 20.27/13.52 | 25.55/17.08 | 10.90/7.15 |
| ✗ | Ours | 64.71/**66.12** | 76.09/**77.59** | 58.00/59.17 | **56.93/54.47** | **66.21/63.90** | **52.78/49.08** | **26.31/24.74** | **32.70/30.81** | **15.63/14.12** |

Table 1: Performance comparison on real-world data captured by Realsense/Kinect.

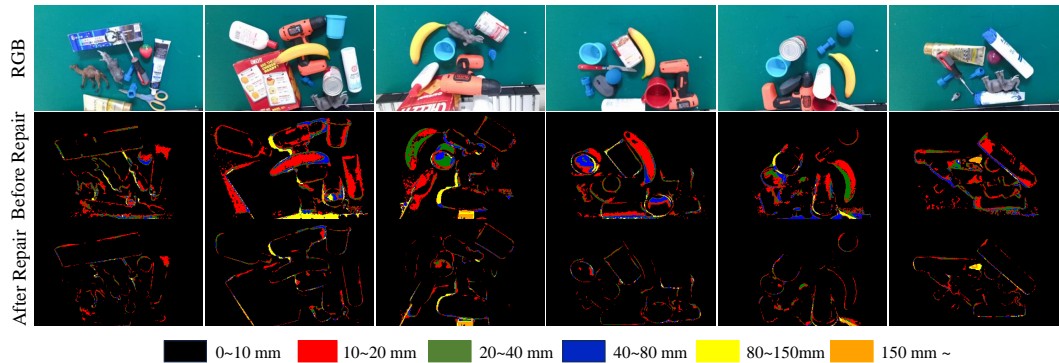

Figure 4: Comparison of camera noise before and after R2SRepairer. Different colors in the noise map represent different noise amplitude ranges, with amplitude measured in millimeters

training solely on R2Sim, there remains a noticeable performance gap compared to the control group. After integrating R2SRepairer (see Line 3), the grasp performance increases significantly and surpasses the grasp detector trained in real-world data. This performance increase steps from mitigating the camera noise, which can be seen in Figure 4. When further adding R2SEnhancer (see Line 5), we notice an increase in AP across the seen, similar, and novel categories by 3.67, 1.63, and 1.49 respectively. These results indicate that our real-to-sim approach bridges the gap between the simulation and real world, where the R2SRepairer and R2SEnhancer modules achieve real-to-sim adaptation at the data and feature levels respectively.

| R | S | R2SRepairer | R2SEnhancer | Seen | | | Similar | | | Novel | | |
|---|---|---|---|---|---|---|---|---|---|---|---|---|
| | | | | AP | $AP_{0.8}$ | $AP_{0.4}$ | AP | $AP_{0.8}$ | $AP_{0.4}$ | AP | $AP_{0.8}$ | $AP_{0.4}$ |
| ✓ | | | | 61.19 | 71.46 | 56.04 | 47.39 | 56.78 | 40.43 | 19.01 | 23.73 | 10.60 |
| | ✓ | | | 46.72 | 55.67 | 38.92 | 46.85 | 56.41 | 39.22 | 14.60 | 18.58 | 7.39 |
| | ✓ | ✓ | | 62.45 | 74.67 | 53.53 | 52.84 | 63.58 | 45.40 | 23.25 | 29.02 | 12.50 |
| | ✓ | | ✓ | 47.77 | 55.92 | 41.45 | 47.96 | 56.02 | 42.42 | 15.76 | 19.95 | 8.60 |
| | ✓ | ✓ | ✓ | **66.12** | **77.59** | **59.17** | **54.47** | **63.90** | **49.08** | **24.74** | **30.81** | **14.12** |

Table 2: Ablation study on two components. The table shows the results on data captured by Kinect. 'R' and 'S' mean the real-world training data and simulated training data respectively.

**Advantages of expandable simulated data.** To verify the advantages of expandable simulated data, we train R2SGrasp on different number of scenes selected from R2Sim and test on real-world dataset as shown in Figure 5. As the number of simulated scenes increases from 100 to 500, the top-1, top-10, and top-50 grasp poses in all test scenarios increase by 6.24, 5.51, and 5.14 AP respectively. The results highlight the benefits of simulated datasets, as we can easily use open-source object meshes in simulated environment to scale up the simulated data for training, thereby continuously improving the performance of the grasp detector.

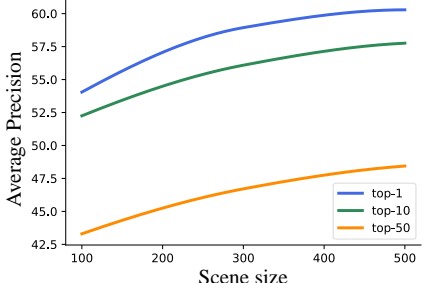

Figure 5: Experiments with different number of scenes. Top-$N$ represents the $N$ grasp poses with the highest scores.

**Memory bank size in R2SEnhancer.** Table 3 illustrates the effect of memory bank size adjustment on grasping performance. As the value of $K$ increases, the grasp performance improves at the

beginning, while reducing after $K$ reaches 120. We believe that a large value of $K$ means more structural features are stored in memory bank, but too large $K$ leads to information redundancy. We experimentally find that the optimal value of $K$ is 120.

| $K$ | all | seen | similar | novel |
|---|---|---|---|---|
| 150 | 47.93 | 65.32 | 54.34 | 24.12 |
| **120** | **48.44** | **66.12** | **54.47** | 24.74 |
| 90 | 48.22 | 65.79 | 53.97 | **24.90** |
| 60 | 47.08 | 64.04 | 53.20 | 24.00 |
| 30 | 45.78 | 63.63 | 50.60 | 23.11 |

| ObjectIDs | GSNet [8] (real) | | ADD-Noise [6] (sim) | | Ours (sim) | |
|---|---|---|---|---|---|---|
| | Attempt | SR | Attempt | SR | Attempt | SR |
| 1,16,19,22,25,31 | 8 | 75% | 6 | 100% | 6 | 100% |
| 8,14,18,30,33,35,36 | 7 | 100% | 8 | 87.5% | 7 | 100% |
| 10,15,17,20,24,28,32,38 | 10 | 80% | 12 | 66.7% | 8 | 100% |
| 3,4,6,11,21,26,37 | 9 | 77.78% | 8 | 87.5% | 8 | 87.5% |
| 5,9,13,20,27,29,39 | 8 | 87.5% | 7 | 100% | 7 | 100% |
| 2,7,12,16,22,23,25,34 | 10 | 80% | 10 | 80% | 8 | 100% |
| Total | 52 | 82.69% | 44 | 84.31% | 43 | 95% |

Table 3: Experiment on memory bank size adjustment.

Table 4: Results of grasping experiment in real world. "real" and "sim" mean that training in real-world data and simulated data respectively.

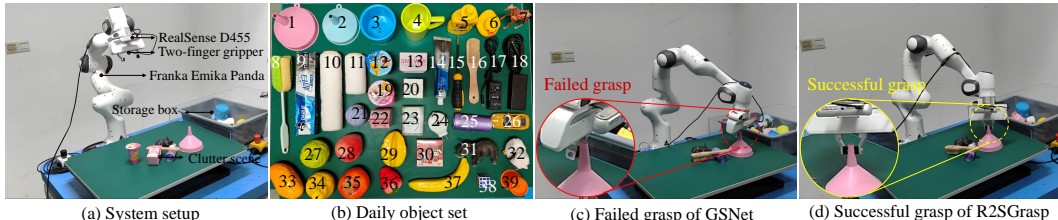

(a) System setup     (b) Daily object set     (c) Failed grasp of GSNet     (d) Successful grasp of R2SGrasp

Figure 6: Description of real-world experiments. (a) shows the experimental setup and environment. (b) displays the objects used in the experiments. (c) and (d) show the failed grasp of GSNet and the successful grasp of R2SGrasp on the same object in the same scenario. Zoom in for better view.

## 5.3 Real Grasping Experiments

To further verify the effectiveness and generalization ability of R2SGrasp, we conduct real-world experiments on the Franka Emika Panda robotic arm with an Intel RealSense D455 camera and a two-finger gripper as illustrated in Figure 6. The experiments are performed on six cluttered scenes, each containing 6-8 randomly placed daily objects. We execute the best grasp pose predicted by grasp detectors for each attempt until all the objects are taken away. The grasping capability is measured by the success rate (SR), defined as the proportion of successful grasps to the total number of grasp attempts. We compare the performance of the grasp detectors [8] trained on GraspNet-1Billion dataset [5], R2Sim dataset with added noise [6] and R2Sim dataset with our method, as shown in Table 4. We find that the grasp detector trained on real data has a lower success rate than grasp detectors trained in simulated data, due to the insufficient amount of real data, which includes only 40 objects, limiting the generalization ability. Using large-scale simulation data improves the grasping success rate, and the success rate can further increase with R2SGrasp. This indicates that our method can effectively leverage grasping skills learned from simulation data in the real world.

## 6 Conclusion & Limitations

**Conclusion.** In this work, we introduce the R2SGrasp framework from a real-to-sim perspective to bridge the gap between simulation and real world in 6-DoF grasp detection. In our framework, we train a robust grasp detector in simulated data, and adapt real world to simulation at both data and feature levels to utilize the skill learned from simulation. Moreover, a large-scale simulated dataset is constructed cost-effectively to enhance the generalization of our methods. By large-scale training and real-to-sim adaptation, the real world generalization ability actually gains great improvement.

**Limitations.** Grasping flat objects or complex shaped objects is suboptimal, and it is a significant challenge in the current 6-DoF grasping community. We think that our real-to-sim idea can help address this issue by generating data in simulated environments that include these types of objects, allowing the network to learn how to grasp them. In the future, we also plan to collect these objects and expand the dataset size to further tackle these challenges.

**Acknowledgments**

This work was supported partially by NSFC(92470202, U21A20471), Guangdong NSF Project (No. 2023B1515040025).

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

# A Appendix

## A.1 More experiments

**Further demonstration of the advantages of real-to-sim way.** Our R2SRepairer can mitigate camera noise which is beneficial for the grasp detector. To further validate our real-to-sim perspective, we also conduct experiment that adding R2SRepairer

| Method | All | Seen | Similar | Novel |
|---|---|---|---|---|
| GSNet [8] w/o R2SRepairer | 42.53 | 61.19 | 47.39 | 19.01 |
| GSNet [8] w R2SRepairer | 43.05 | 61.39 | 48.05 | 19.72 |
| Ours | 48.44 | 66.12 | 54.47 | 24.74 |

Table S1: Average precision comparison on real-world data captured by Kinect.

to the grasp detector [8] trained on real-world data. As shown in Table S1, after adding R2SRepairer (see Line 2), the grasp performance shows a modest improvement, but it still lags significantly behind our method that is trained on simulated data. This demonstrates that a large amount of simulated data can train a more robust grasp detector, and by utilizing our real-to-sim method, this capability can be effectively applied to real-world data.

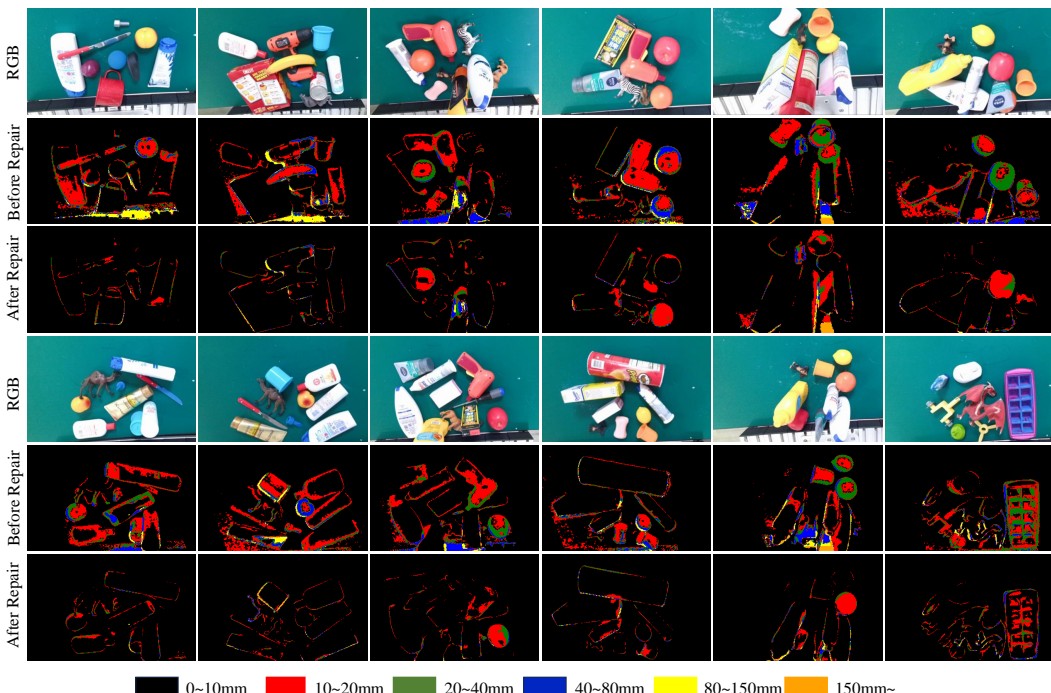

Figure S1: Comparison of camera noise before and after R2SRepairer. Different colors in the noise map represent different noise amplitude ranges, with amplitude measured in millimeters

**Quantitative analysis of R2SRepairer.** We use the depth maps from GraspNet-1Billion [5] test set to test R2SRepairer, which consists of 22950 samples. The evaluation metric is the root-mean-square

| Prediction | Input | Output | RMSE(mm) ↓ |
|---|---|---|---|
| Depth Anything V2 [30] | RGB | Depth value | 206 |
| Ours | RGB | Depth value | 220 |
| Ours | RGBD | Depth value | 8.55 |
| Ours | RGBD | Depth residual value | 7.91 |

Table S2: Comparison of depth repair Performance.

error (RMSE) of the predicted depth map, measured in millimeter. The results showed in Table S2 indicate that if the network only deals with RGB images, the depth value prediction is not accurate, which reduces the grasp performance. Moreover, predicting the residual values is more effective than predicting the depth values.

**Qualitative analysis of R2SRepairer.** We futher demonstrate the effectiveness of R2SRepairer by presenting camera noise before and after refinement. As shown in Figure S1, it is evident that

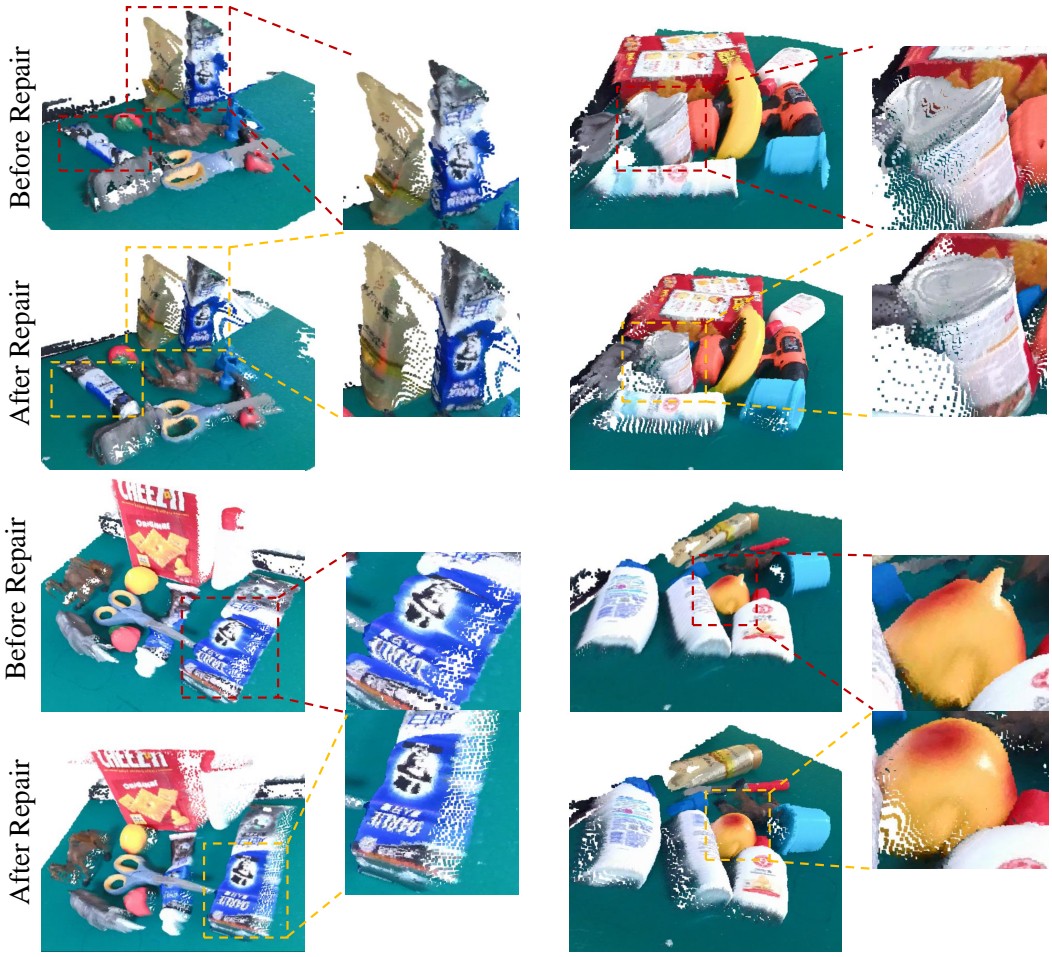

Figure S2: Visualization of single-view point cloud before and after repair. Zoom in better view.

the correct noise image significantly reduce noise compared to the initial noise image. To further demonstrate the performance of R2SRepairer, we compared the single-view point cloud before and after camera noise repair. As shown in Figure S2, after noise repair, the positional drift and structural deformation of the point cloud are mitigated, bridging the gap between real and simulated data.

**The impact of noise in Residual labels.** R2SRepairer is robust to the noise in residual labels. As shown in Table S3, we add Gaussian noise with a mean of 0 and different variance (std) to the residual labels. The overall AP value increases after adding Gaussian noise, with a more significant increase when std=10. We believe that introducing appropriate noise can enhance the robustness of model learning.

| Std | All | Seen | Similar | Novel |
|-----|-------|-------|---------|-------|
| 0 | 48.44 | 66.12 | 54.47 | 24.74 |
| 5 | 48.84 | 67.29 | 54.53 | 24.69 |
| 10 | 49.67 | 67.81 | 55.93 | 25.25 |

Table S3: Experiment on noise adjustment in residual labels.

**Qualitative analysis of structural features in memory bank.** Our Real-to-Sim Feature Enhancer (R2SEnhancer) uses the precise structural features stored in the memory bank to enhance the real-world features. To visually demonstrate the semantic information of the stored structural features, we visualize the structures represented by these features. First, we calculate the cosine distance between the features of each graspable point and the stored features in the memory bank, assigning each graspable point to the nearest stored structural feature. Then, we obtain a set of graspable points for each stored structural feature. Based on cosine similarity, we select three distinct features from the memory bank and visualize their corresponding sets of graspable points. As depicted in

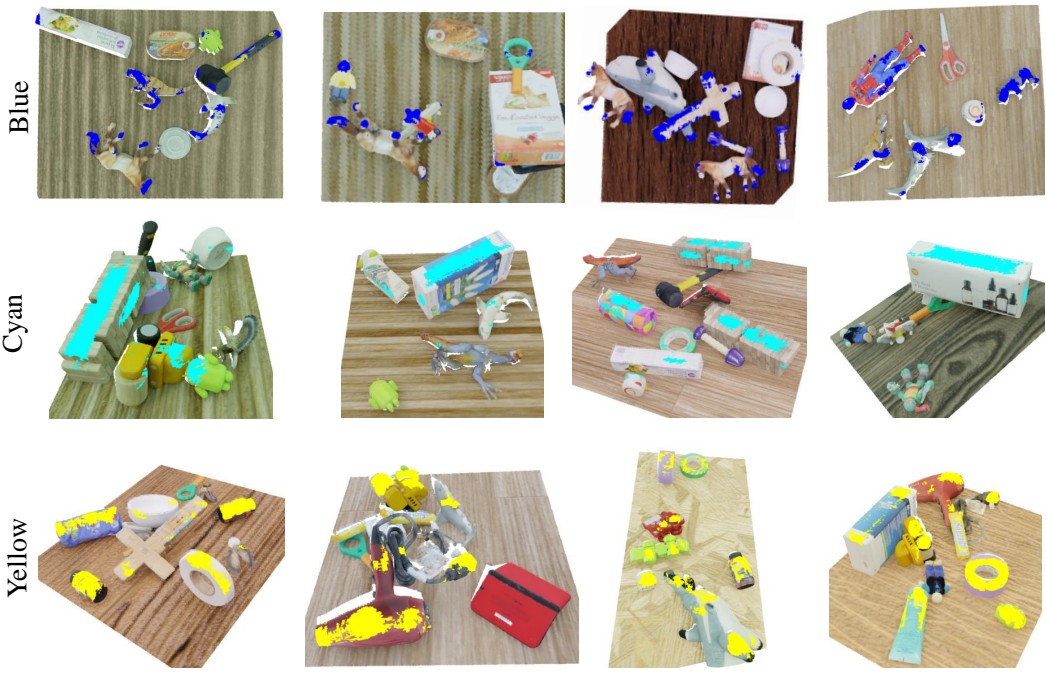

Figure S3: Semantic information of the stored structural features. "Blue", "Cyan", "Yellow" represent the structures corresponding to the three selected structural features.

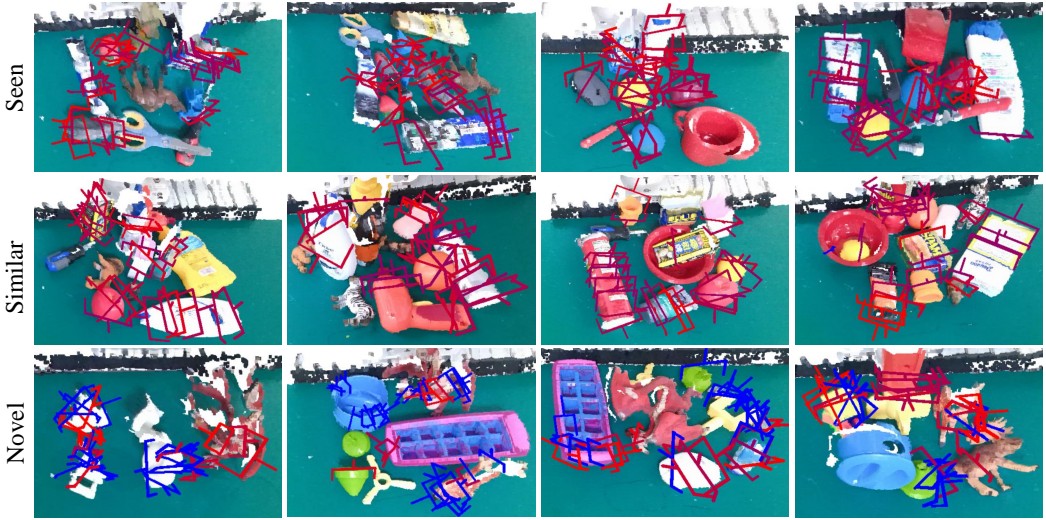

Figure S4: Top-30 grasp poses predicted by R2SGrasp on the test set of GraspNet-1Billion. The red gripper indicates successful grasp pose, while the blue gripper indicates failed grasp pose.

Figure S3, there are obvious differences in the structure represented by the three features. The first feature represents sharp object structures, as shown in the first row of Figure S3, which are commonly found on toy legs and heads. The second feature represents planar object structures, as depicted in the second row, primarily seen on square boxes. The third feature represents curved object structures, as illustrated in the last row, appearing on various curved surfaces, with bottles being the most prominent.

**Qualitative analysis of grasping performance.** To demonstrate that our R2SGrasp can adapt to real data, we use R2SGrasp to predict grasp poses on the GraspNet-1Billion test set and visualize the results, as shown in Figure S4. In seen and similar scenes, R2SGrasp predicts grasping poses

with a success rate close to 100%. In novel scenes, there are some failed cases, which occur due to collisions or grasping at empty locations.

## A.2  Implementation details of grasp detector.

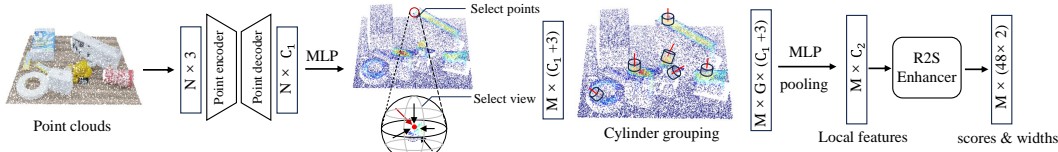

Figure S5: Grasp detector in details

**Architecture design of grasp detector.** The grasp detector in details is shown in Figure S5. We first randomly sample N points from the single-view point clouds generated from depth map, and then use a point cloud backbone to extract point-wise features with $C_1$ dimension. The point cloud backbone adopt a Unet [27] architecture with a ResNet14 [28] encoder built upon the Minkowski Engine [31]. Followed by a MLP layer, we predict the object point mask $I_o$ and graspness heatmap $I_h$ to select the graspable points along with their corresponding point-wise features with the shape of $M \times (C_1 + 3)$, where M is the number of graspable points and 3 denotes the cartesian coordinates of the points. We also select the grasp view of the graspable points from the predefined 300 approaching vectors based on the grasp view scores $s_v$ which is also predicted by a MLP layer. Then, we perform cylinder grouping operation along the grasp view for each grasp point to aggregate the features of G neighboring points, followed by the MLP and max pooling operations to extract local structural features with $C_2$ dimension. Finally, our Real-to-Sim Feature Enhancer (R2SEnhancer) refines the local structural features using the stored simulated features and outputs the grasp scores $s_g$ and widths $s_w$ shaped as $M \times 48$, where 48 denotes the grasp candidates of the grasp points. For our network, we set $N = 20000, M = 1024, C_1 = 512, C_2 = 256, G = 16$.

**Loss Design.** The grasp detector is trained with the following loss function:

$$L_g = L_o(I_o, I_o^*) + \lambda_1 L_h(I_h, I_h^*) + \lambda_2 L_v(s_v, s_v^*) + \lambda_3 L_s(s_g, s_g^*) + \lambda_4 L_w(s_w, s_w^*), \quad (5)$$

where $L_o, L_h, L_v, L_s, L_w$ are used to supervise the learning of object points, graspable points, grasp views, grasp scores and grasp widths respectively. $I_o^*, I_h^*, s_v^*, s_g^*, s_w^*$ is the ground truth of object point mask, graspness heatmap, grasp view scores , grasp scores and grasp widths. $L_o$ adopts binary classification loss, while others use regression loss. Due to the simplification of our grasp annotations, some grasp poses may lack supervision signals. Therefore, when calculating the loss, we ignore any predicted grasp poses that lack supervision signals.

## A.3  R2Sim dataset details

The overall process of dataset construction is shown in Figure S6. We start by selecting 256 daily household objects from the Google Scanned Objects dataset [26] and the GraspNet-1Billion training dataset [5]. Then, we generate scenes in blenderproc [25] and simultaneously label the grasp poses of the objects. After that, we project object-level grasp annotations into the scenes and detect the grasp annotations that result in collisions. Finally, we adopt our proposed grasp annotation simplification method to remove ineffective grasp poses. The remaining grasps serve as scene-level annotations. To sum up, our R2Sim dataset comprises 500 scenes with 76,800 RGB-D images. Each scene contains approximately 14.4 million grasp annotations and each frame in every scene is also annotated with object segmentation maps, 6-DoF poses of objects and camera, graspness heatmap and view graspness. In the graspness heatmap, brighter areas indicate a higher likelihood of successful grasps. Similarly, higher values of view graspness also represent a greater probability of successful grasps. Some examples of RGB, depth map, segmentation map and graspness map are shown in Figure S7.

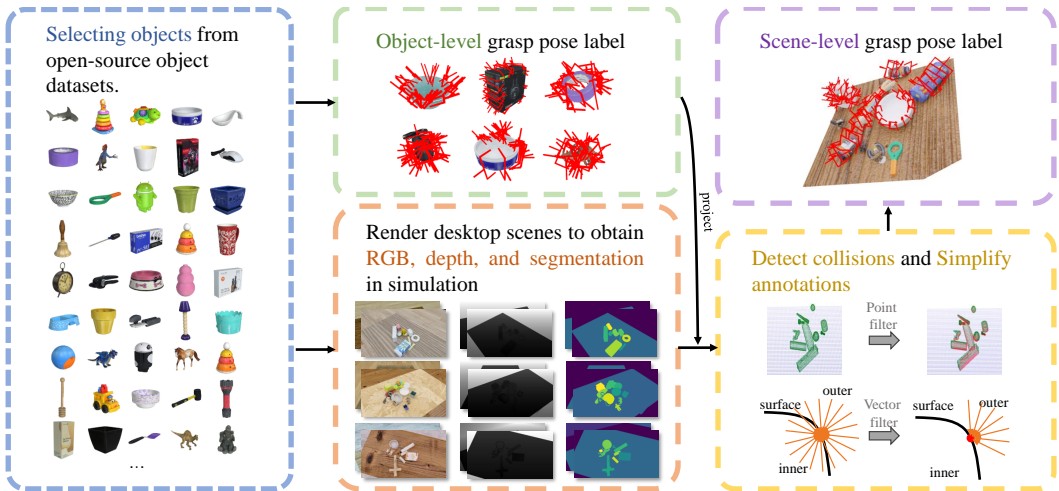

Figure S6: Overview of data generation pipline.

### A.3.1 Details of object level grasp annotation.

We use a sampling-evaluation approach to annotate the grasp poses of objects. Grasp poses are determined by downsampling high-quality mesh models to ensure that the grasp points are evenly distributed in the voxel space. For each grasp points, 300 approach directions are sampled uniformly on a spherical space. Grasp candidates of each approach directions are explored on a grid defined by 4 gripper depths and 12 rotation angles. To sum up, there are 48 grasp candidates along each approach direction and 14,400 grasp candidates on each grasp point. The gripper width is adjusted as necessary to prevent empty grasps or collisions. We adopt analytic computation method as [5? ] to grade the sampled grasp poses. The grasp scores range from 0 to 1, with higher scores indicating a greater likelihood of successful grasps.

### A.3.2 Details of scene level grasp annotation.

Using the object poses, the grasp pose in object coordinate system is projected onto the world coordinate system. We detect collisions for the projected grasp poses in the scene and set the scores of those that collide to zero. Assuming there are N grasp points projected into the scene, we calculate the success rate of grasp candidates at each point, resulting in N graspness values. Using the camera's intrinsic parameters, we convert the depth map into a single-view point cloud. We then use the K-NN algorithm to match the grasp points in the scene, assigning each point in the point cloud the corresponding graspness value. This value is then back-projected into the image to create the graspness heatmap. Similarly, we calculate the success rate of grasp candidates along each approach direction and using this as the view-graspness. We also make further simplification of grasp pose annotation to improve training efficiency.

### A.3.3 Scene generation.

We automated the construction of cluttered desktop scenes using Blenderproc [25]. We first construct a simple indoor scene with a table placed at the center. Textures for the table, floor, and walls are chosen randomly from specific material categories provided on AmbientCG. The lighting setup of the scene is randomized as well, with practical adjustments to intensity and variations in light color to improve visual clarity and accuracy. Then, we choose a variable number of objects ranging from 7 to 10 from our object pool and place on the table. To create sufficiently cluttered scenes, we place the objects 1.5 meters above the table and allow them to fall naturally onto the surface. Finally, we set 128 camera poses to capture RGB-D images from multi views, where the poses are randomly sampled on the upper hemisphere, with a radius of 1.1 meters centered on the objects' region. Based

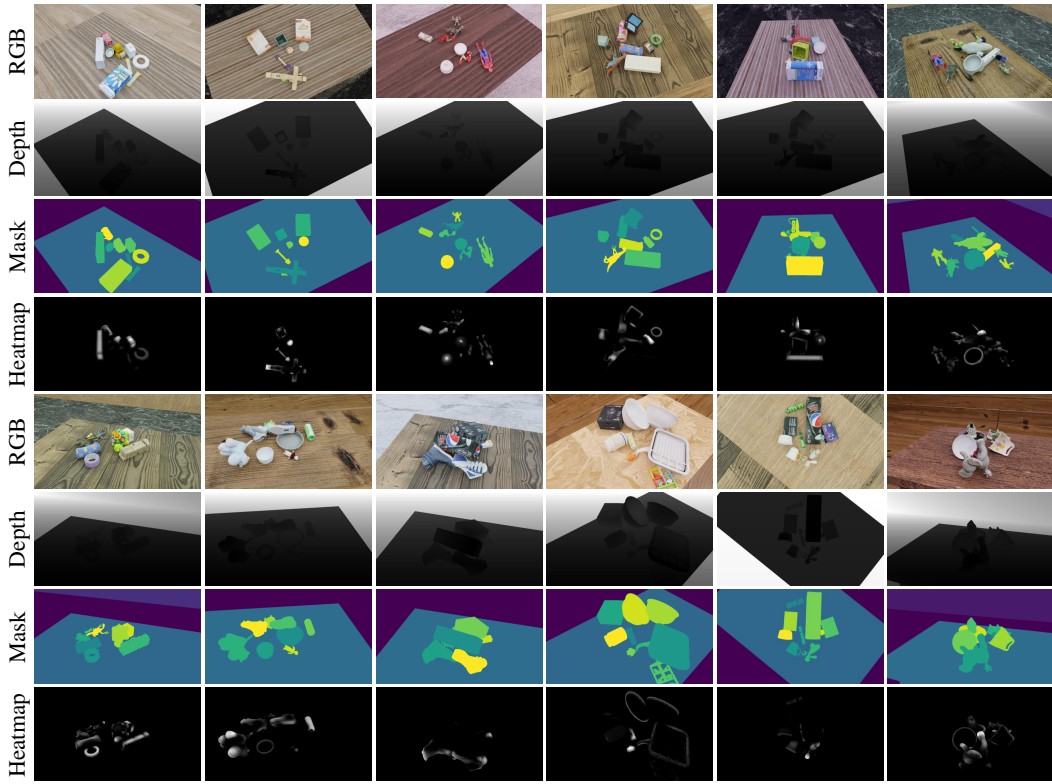

Figure S7: Display of RGB, depth map, segmentation mask and graspness heatmap in our simulated dataset.

on the above setup, we obtain RGB-D images, object segmentation maps, object poses, and camera poses from different angles efficiently.

### A.3.4 Analysis of simplified annotation.

To demonstrate the advantage of simplified annotation, we compare the simplified and non-simplified annotations across multiple metrics. The average precision measures the performance of grasp detector trained on 300 scenes selected from R2Sim dataset. Except for the average accuracy, all other metrics are evaluated on the entire R2Sim

| Metrics | Simplified | Non-Simplified |
|---|---|---|
| Average Precision(%) | 35.94 | 33.68 |
| Run Time(epoch/h) | 5.64 | 21.17 |
| Memory Usage(GB) | 7.50 | 48.41 |
| GPU Memory Usage(GB) | 4.43 | 9.77 |
| Storage Usage(GB) | 15 | 72.44 |

Table S4: Compare metircs between simplified and non-simplified annotations.

dataset. As shown in Table S4, following the simplification of annotations, there are a notable increase of 2.26 AP. We believe that this improvement is due to the simplified annotations alleviating the imbalance between positive and negative samples present in the original annotations, where positive samples made up less than 2% of the total according to statistics. Moreover, after simplifying the annotations, the program's runtime, memory usage, and GPU usage decrease by 73.36%, 84.51%, and 54.66%, respectively, and the storage usage for the entire dataset annotation decrease by 79.29%. This simplified annotation method is crucial for constructing a large-scale dataset, as it reduces the burden of data storage and neural network training.

