# OpenReview forum: "Real-to-Sim Grasp: Rethinking the Gap between Simulation and Real World in Grasp Detection"
_robot-learning.org/CoRL/2024/Conference — CoRL 2024_

### Official Review · Reviewer_xCby · 2024-07-06
**Real-to-sim: a new idea for bridging the gap between real-to-sim generalizations for purely vision based models**

**Originality:** 3
**Technical Quality:** 4
**Clarity Of Presentation:** 4
**Potential Impact:** 3
**Recommendation:** 3
**Confidence:** 3

**Review:**

This paper presented a new idea of bridging the gap between real-to-sim generalizations for purely vision based models. The writing and explanation of ideas were very clear. Sufficient experiments including comparison with baselines, ablation studies and real-world robot hardware demonstration video were all very helpful.

R2SRepairer and R2SEnhancer are the key contributions of this paper. The explanation of R2SEnhancer was clear, however, R2SRepairer training dataset and validation procedure needs more details. Following are some of the questions, that are not clear to me:

1. On lines 134-135, the authors mentioned: “The generation of twin datasets are not difficult since it does not need to construct grasp annotations, which is easy to be adapted to different depth cameras”. However, no further information was provided on how this “twin” dataset was actually generated. From the best of my knowledge, the generation process would look something like this: authors will set up a static scene in real-world and capture its RGB-D images from multi-view calibrated cameras. Then, somehow combine these multi-view images to get noise-less depth maps. I understand that this is easier than getting pose and grasp labels for all the objects in the scene. However, I am not sure, how multi-view images are combined? Also, I can imagine that multi-view images will help reduce the noise, but not sure how it will help with the positional drift?

2. In experiment section, on line 188-190, the authors provided some details about dataset for R2SRepairer: “use twin datasets based on GraspNet189 1Billion [5] to train our R2SRepairer. The grasping performance is validated using the test set 190 from GraspNet-1Billion”. Does this mean that R2SRepairer was trained on 100 real-world train scenes from the said dataset? How were the ground-truth depth-maps obtained? Also, what was the performance of this module on test set?


Minor Typo on L112,  “Expect the two modules” should be “Except the two modules”.

**Quality Of The Limitations Section:**

1

**Questions For Rebuttal:**

Please provide more details of training and testing dataset for R2SRepairer. Specifically, how this dataset was collected/obtained? What was the train and test set size? What was the performance on test set?

**Robotics Focus:**

4

**Summary Of Paper:**

In this work, authors proposed a new way to train 6DoF grasp detection models. Real-world annotated grasp dataset is usually very limited in size, as it is costly to label such dataset, and does not scale well. Traditionally, research works have tried to address this issue either via training on noisy simulated data, or via reducing noise in real world data via extrapolating/averaging or smoothing. Authors show that training on noisy data has adverse effect on model performance (Figure 1). Instead, authors propose a new idea of Real-to-Sim (R2S) grasp framework, where they train on noise-less simulated data, and during deployement/inference, enrich the real-world data with (a) R2S-Repairer: reducing the noise and positional drfit, and (b) R2S-Enhancer: enhance the real-world features via matching them with a dataset of simulated features. Via extensive ablation studies, comparison with baselines, and real-world robot demonstrations, authors show that their method performs close to / superior than models trained only with simulated data or even real data.

**Summary Of Recommendation:**

Authors present a new approach for training grasp detection models in sim, and transforming real-world to be close to sim. The approach seems to work well. More details on datasets generation for R2SRepairer would be very helpful.

---

### Official Review · Reviewer_SR4p · 2024-07-20
**Grasp Predictor Method that is Trained in Sim.**

**Originality:** 3
**Technical Quality:** 3
**Clarity Of Presentation:** 3
**Potential Impact:** 3
**Recommendation:** 3
**Confidence:** 5

**Review:**

- The method uses Rs2Repairer to improve the quality of depth map and R2SEnhancer to improve the features. The enhancer model basically learns a memory bank during training and at inference time it encodes the local features and enhances the features through cross attention between local features and the memory bank. Majority of benefits come from the Rs2Repairer and the enhancer has marginal effect.

- It seems like Rs2Repairer requires real data + reconstructed meshes are required to train the repairer. Please clarify what is the training data used for Rs2Repairer.

- Missing Comparison: The paper claims that it is the state-of-the-art method for grasp prediction that uses simulation data. Most of the other methods are related to graspnet 1-billion. I think a comparison with M2T2 [Yuan et al, CoRL 2023] as that method is also only trained with synthetic data and transfers to the real world makes a lot of sense. It would strengthen the paper.

- Overclaiming concern: The paper claims that it is the first method that only uses sim data and beating real world methods. However there is a long history of methods using point clouds or 2.5 depth images (e.g. DexNet series) that even led to starting robotic bin picking companies.

- What are the limitations of the method? Where does it fail?

**Quality Of The Limitations Section:**

1

**Questions For Rebuttal:**

See Above.

**Robotics Focus:**

4

**Summary Of Paper:**

A method for predicting 6-DoF grasps of objects through learning a depth map denoiser to predict the residual depth maps. The method also trains a feature enhancer to improve the quality of features to predict the 6-DoF grasps but that has marginal effect.

**Summary Of Recommendation:**

I improved my rating after rebuttal.

---

### Official Review · Reviewer_TS41 · 2024-07-21
**A novel denoising approach to closing the sim-to-real gap for grasp detection**

**Originality:** 3
**Technical Quality:** 3
**Clarity Of Presentation:** 3
**Potential Impact:** 3
**Recommendation:** 4
**Confidence:** 3

**Review:**

**Summary:** The authors consider the problem of transferring grasp policies from simulation to reality. Typically, this is addressed by adding noise to the simulation data that policies are trained over (i.e., domain randomization), but the authors point out this approach is heuristic, and actually reduces the performance of the resulting policy. In contrast, they propose an approach where they train a policy over noiseless data in simulation, and online, pass the noisy sensor data through a trained pipeline to denoise it before passing it to the policy. The authors support their claims with good-quality hardware experiments, demonstrating both the performance drop induced by domain randomization, and successful sim-to-real transfer using their learned denoiser.

**Strengths:**

- Very interesting perspective + idea: instead of noising simulation data, denoise real data. Trains models to denoise real-world image data (using digital twins constructed in Blender) along with grasp feature detector that can rely on 3D primitives accessible in simulation.
- Strong results: show their method outperforms methods trained only on real data via access to simulation, and demonstrate compelling sim-to-real performance.
- Thorough real-world experiments with enough data to make a compelling sim-to-real case.
- Good comparisons against existing methods for grasp detection (GraspNet1B) and a more traditional domain randomization-based approach. Hardware experiments show a clear improvement from applying the proposed method.
- Mostly clearly written + a simple, but compelling story.

**Weaknesses:**

- Missing an explicit discussion of paper limitations (required by CoRL guidelines).
- Hardware experiments are also somewhat limited in scope (36 objects, which is good, but grasped only once per method) — more data + object diversity could better support the authors’ claims.
- Parts of the methods section need editing for clarity. In particular, the actual training process for the depth reconstruction is unclear — the authors say simply they generate a “digital twin dataset” and learn a residual between the simulated point clouds (treated as ground truth) and real ones. The “digital twin dataset” seems like a core contribution, and more explanation is needed (either in the paper or the appendix) to better understand the core insights behind this process.
- Unclear if the problem of grasp detection is still compelling -- reported success rates are similar to or below those reported by DexNet and other techniques from 10+ years ago. Applying this approach to other, more contact-rich problems, if the benefits of real-to-sim were still present, would provide a more compelling and exciting demonstration of the core idea.

**Summary after Rebuttal Phase:** Overall, I think this paper contains some nice, novel ideas re: real-to-sim and thinks seriously about getting grasp generation to work well with real perception stacks in hardware. I want to commend the authors on performing a DexNet baseline (something I haven't seen in years of reviewing 6DoF grasp papers) -- while they don't significantly outperform it, they *do* outperform it, and clearly beat other 6DoF baselines. Between the improved performance + new baseline + nice algorithmic contributions I will bump my score to a strong accept.

**Quality Of The Limitations Section:**

1

**Questions For Rebuttal:**

- What are some core limitations of the proposed approach?
- How do you collect the "digital twin" data needed to train the proposed denoiser?
- If the residual labels are themselves noisy, does this create problems for the method?
- How much calibration data is needed?
- Do you think the results would hold if only working with RGB images? Can you hypothesize re: how important depth is to the core setup?
- Why is grasp detection a "hard" problem? Is your method (or the other grasp detection methods you considered) able to synthesize grasps for objects, or perform better, than a system like DexNet [1], which reported 95% success rates in large-scale, real-world trials?

[1] Dex-Net 2.0: Deep Learning to Plan Robust Grasps with Synthetic Point Clouds and Analytic Grasp Metrics, Mahler et al., RSS '17.

n.b.: My apologies if the last question comes across as pointed -- I would not push to reject your paper over something like this, I am just curious why this subfield remains quite active after an impressive result like DexNet + mid-2010s methods are not compared to or cited within this community.

**Robotics Focus:**

4

**Summary Of Paper:**

Learning to denoise raw sensor data to match the "clean" inputs seen during simulation training can enable effective and non-conservative sim-to-real transfer.

**Summary Of Recommendation:**

The particular approach to real-to-sim here (denoising hardware data to match simulation data, to prevent dist. shift) is quite interesting. While the paper is not perfect, I think this algorithmic improvement is of interest to the broader CoRL community.

---

### Official Review · Reviewer_7Q2J · 2024-07-22
**Paper Review for Real-to-Sim Grasp: Rethinking the Gap between Simulation and Real World in Grasp Detection Download PDF**

**Originality:** 2
**Technical Quality:** 3
**Clarity Of Presentation:** 3
**Potential Impact:** 2
**Recommendation:** 2
**Confidence:** 3

**Review:**

**Strengths and weaknesses**

**Strengths**
1. This paper proposes a real-to-sim approach, which is different from common sim-to-real ones, that bridges the perception gaps between sim and real and enjoys better generalization.
2. The contribution of the R2Sim dataset, with extensive annotations and diverse scenarios, provides a robust training ground for the community.

**Weaknesses**
1. The gap between sim and real is usually two-fold, dynamics and perception. It seems that this paper only discusses and addresses the perception gap. The framework's performance is contingent on the physical plausibility of the simulated data, and inaccuracies in the simulation could impact the real-world applicability. From my own experience, when deploying a grasping policy from sim to real, a lot of failures are actually due to the dynamics mismatch between sim and real, such as simulation artifacts, dynamics friction etc. Failure to mention and address the dynamics mismatch somewhat weakens the paper.
2. There is no limitation section.

**Quality Of The Limitations Section:**

1

**Questions For Rebuttal:**

1. What are the computational requirements for training and deploying R2SGrasp on a real robot? How does the performance compare to other methods in terms of processing time and resource usage?

2. How robust is the proposed method in real-time applications with dynamic environments? Some discussion around this will be appreciated.

3. It would be interesting to see the common failure modes of the proposed methods, such as unreachable joint position, gripper collision, bad repaired depth, etc. Some discussions and analysis will make the paper stronger.

**Robotics Focus:**

4

**Summary Of Paper:**

The paper presents a novel framework, R2SGrasp, aimed at improving 6-DoF grasp detection by addressing the gap between simulation and real-world data. Unlike traditional sim-to-real approaches, which adapt simulated data to real-world noise, R2SGrasp employs a real-to-sim method. Specifically, this paper uses R2SRepairer to mitigate camera noise in real depth maps (data level) and R2SEnhancer to enhance real features with simulated geometric primitives (feature level). In addition, this paper proposes a large-scale simulated dataset, R2Sim, containing 64000 RGB-D images and 14.4 million grasp annotations. Experimental results demonstrate that R2SGrasp outperforms other baseline methods that are trained on real-world data and sim-to-real methods.

**Summary Of Recommendation:**

I recommend weak reject of this paper based on the aforementioned weaknesses. I am happy to raise my score if the authors provide clarifications on the questions I raised and address my concerns in the review.

---

### Author Rebuttal · Authors · 2024-08-10

We extend our gratitude to reviewers for their careful reading, thoughtful feedback, and valuable insights! Based on the reviewers' comments and suggestions, we summarize the strengths of our work:

1. **A novel real-to-sim perspective** (7Q2J: "different from common sim-to-real ones", TS41: "Very interesting perspective + idea: instead of noising simulation data, denoise real data", xCby: "This paper presented a new idea of bridging the gap between real-to-sim generalizations for purely vision based models.")
2. **Sufficient expriments and strong results** (TS41: "Strong results: show their method outperforms methods", "Thorough real-world experiments with enough data to make a compelling sim-to-real case.", xCby: "Sufficient experiments including comparison with baselines, ablation studies and real-world robot hardware demonstration video were all very helpful.")
3. **Provide a diverse dataset** (7Q2J: "provides a robust training ground for the community.")
4. **Clear writing** (TS41: "Mostly clearly written + a simple", xCby: "The writing and explanation of ideas were very clear.")

According to the feedback of reviews, we conduct additional experiments and analysis to further improve our work in rebuttal.

1. We compare with more other methods to demonstrate our superior performance, robustness, and generalizability.
2. We provide a detailed explanation of our training data and training process.
3. We add details about the limitations of our method.

---

### Decision · Program_Chairs · 2024-09-04

**Decision:**

Accept

**Comment:**

The paper presents a novel framework, R2SGrasp, aimed at improving 6-DoF grasp detection by addressing the gap between simulation and real-world data. R2SGrasp employs a real-to-sim method comprising of R2SRepairer to mitigate camera noise in real depth maps and R2SEnhancer to enhance real features with simulated geometric primitives. The paper also proposes a large-scale simulated dataset, R2Sim, containing 64000 RGB-D images and 14.4 million grasp annotations.

Reviewers favorably viewed the proposed method and the experimental results. However, the reviewers also raised some concerns: applicability of proposed approach to other more contact rich tasks, comparisons to other related works, some concerns about claims made in the paper, and the lack of many important training details.

The author response addressed many reviewer concerns. `TS41`, `xCby`, `SR4p` reviewed the author response and settled on accept leaning ratings. The AC looked through the paper, the reviews, and the author response. While the residual concern from `7Q2J` (gap in dynamics) is well placed the AC agrees with `TS41` that it may be beyond the scope of the current paper. The authors are encouraged to include the author-reviewer discussion when revising the paper, particularly the points about additional experiments, appropriate framing of claims, and adding a discussion of the limitations.